# Bioinspiration in Fashion—A Review

**DOI:** 10.3390/biomimetics4010016

**Published:** 2019-02-12

**Authors:** Jane Wood

**Affiliations:** Manchester Fashion Institute, Manchester Metropolitan University, Manchester M15 6BH, UK; j.e.wood@mmu.ac.uk

**Keywords:** sustainability, biomimicry, fashion, apparel, bacterial cellulose, mycelium, textiles

## Abstract

This paper provides an overview of the main technologies currently being investigated in the textile industry as alternatives to contemporary fashion fabrics. The present status of the textile industry and its impact on the environment is discussed, and the key drivers for change are highlighted. Historical use of bioinspiration in synthetic textiles is evaluated, with the impact of these developments on the fashion and apparel industries described. The review then discusses the move to nature as a supplier of new fabric sources with several alternatives explored, drawing special attention to the sustainability and performance aspects of these new sources.

## 1. Introduction

The unsustainable rate of consumption attributed to the fashion industry has been well documented [1]. It is estimated that £140 million worth of clothing goes into landfills each year, with 24% of consumers surveyed stating they had disposed of clothing after only one wear [2]. The Waste and Resources Action Plan (WRAP), Valuing Our Clothes, report suggests that the annual footprint of a household’s newly bought clothing, along with the washing and cleaning of its clothes, is estimated to be equivalent to the carbon emissions from driving an average modern car for 6000 miles and the water needed to fill over 1000 bathtubs [2]. In addition to this, there is waste hidden in the supply chain with estimates that 15% of fabric is rejected and discarded before it leaves the factory [3]. Whilst large volumes of synthetic fabrics and garments find their way to landfill, the environmental impact of natural fiber production cannot be ignored. Cotton, considered by many to be eco-friendly, requires large volumes of chemicals to ensure production volumes meet demand; this has devastated some regions, such as the Aral Sea in Central Asia. Once a rich and fertile land, this region of the world has become near desert-like due to the overfarming of cotton.

These facts paint a bleak picture and suggest the textile industry is in crisis. With added consumer awareness around sustainability issues, the need for change is high on the agenda.

This review will discuss how the textile industry has historically used inspiration from nature to manipulate man-made products and how it is now moving into a new era, looking at nature as a source of new materials.

## 2. Inspiration from Nature

### 2.1. Velcro

Velcro^®^ is possibly the most well-known example of biomimicry in textiles [4,5]. In 1941, the Swiss inventor George de Mestral noticed that his dog’s fur was covered in small burdock plant burrs after walking in the fields. Upon closer inspection, Mestral discovered that the burrs were covered in small hooked spines, allowing them to attach to other materials [6]. Further investigation led to the development of the hook and loop tape that continues to have numerous uses across the whole textile industry. This type of development is the perfect example of true inspiration from nature and how a simple investigation of the natural world can be translated into product, which continues to have a huge impact in a variety of end-uses.

### 2.2. Pinecones

The current trend in technical textiles is smart fabrics, those that can sense, adapt, and react to their surroundings [7]. While most of this effect is created on a polymeric scale, some of the principles are based on those found in nature.

The structure of a pinecone is such that seeds are only released once optimal conditions for germination are sensed. In nonoptimal conditions, the pinecone structure remains tightly closed, protecting the seeds. However, at appropriate temperature and humidity levels, the bract scales of the cone bend. This bending is caused by differential swelling of the two types of cellulose found in the bract scale [4,5,8]. The opening of the cone allows seed dispersion and the potential for germination.

This principle has been adopted in both fiber/fabric structures and chemical finishes. Schoeller Textil AG (Sevelen, Switzerland), a Swiss technical textile manufacturer, states that the opening and closing of the pine cone structure was used as inspiration for the development of their c_change fabric [9]. This fabric is already manufactured in bulk and can be found in premium range sports garments [10]. It consists of a polymer membrane sandwiched between two textile layers. When the body is warm, the molecular structure of the polymeric membrane opens to allow the escape of excess heat and moisture. As the body cools, the structure contracts, increasing the insulating properties of the fabric, thus trapping heat against the skin [9]. The company claims that this fabric ensures an optimal body climate [11] due to the adaptive nature of the textile. 

MMT Textiles Ltd. (London, UK) has used the reaction to atmospheric conditions of the bract scales found in pine cones as inspiration for the development of Inotek^TM^ fibers [12]. Using the differential swelling principle of pine cone cellulose as a starting point, Inotek^TM^ fibers are engineered to curl and become shorter in length as atmospheric humidity increases. Due to the placement of the fibers in relation to the yarn surface, the yarn becomes thinner as the humidity increases, effectively “opening” the structure. Therefore, the fiber becomes more permeable to air, enabling the flow of moist warm air away from the body and allowing a comfortable temperature to be maintained [5]. As humidity decreases, the fibers return to their original state, thus closing the structure and imparting a degree of insulation to the body. Yet to be found in bulk production, MMT Textiles Ltd. states that they are working with a number of global industry partners to bring this product to the wider market, and that the fiber can be easily spun into yarns to suit both knitted and woven fabric structures [12].

### 2.3. Sharkskin

While many innovations have focused on the surface structure of the textile, the development of the Fastskin suit (Speedo International Ltd., Nottingham, UK) considered the whole garment and how the shape of the animal itself could influence this.

Initially, the skin of the shark was studied with special attention given to the dermal denticles (scale-like structures) on the surface [13]. The denticles are arranged longitudinally on the body axis and in line with the flow of water, but their spacing varies depending on their location on the body [5,14]. It is understood that this arrangement minimizes friction when moving in the water [15]. Developers mimicked this denticle arrangement on the surface of the fabric using a knitted, ribbed structure [16], with tests showing that water friction was reduced by 7.5% [17]. However, it is not only the dermal denticles that contribute to the hydrodynamics of a shark’s movement, the shape of the body itself is also a key contributor. Therefore, the swimsuit was engineered to give compression in certain areas, effectively changing the shape of the swimmer’s body to further decrease water drag. The seams of the garment were bonded (rather than stitched) to enhance the fit of the garment, allowing the seams to lie flatter against the body [18]. It was rumored that the suit took around 25 min to put on, and swimmers needed to be cut out of the garments due to the closeness of the fit. Such was the improvement in speed of the swimmers through the water, over 300 records were set in the 2008–2009 season. This led to questions being raised by the International Swimming Federation (FINA), the governing body for competitive swimming, and ultimately a ban on these suits being used in competitions [5].

### 2.4. Stomatex

The mechanism of respiration in plants requires gaseous exchange of oxygen, carbon dioxide, and water vapor. Stomata (tiny pores) on the surface of plant leaves allow this process to occur. The stomata are thought to open in daylight and close during the hours of darkness, with the motion being controlled by guard cells that react to internal pressure within the plant structure [5].

Stomatex^®^ is a composite technical textile [19,20] that is based on these principles. The base textile is neoprene (a synthetic polymer foam rubber) encapsulated within a synthetic knitted outer layer, into which small dome-shaped structures have been embossed. Each dome has a small hole (pore) at its apex, as illustrated in Figure 1. The textile is designed to be close-fitting, thus able to react to the body’s movements. At rest, any excess heat and moisture rises into the domes and is released via the pore. When the body is moving, the domes (and pores) flex and move, allowing heat and moisture out, and cooler air in, thus maintaining a comfortable microclimate [11]. Stomatex^®^ has found specific applications in garments for athletes, particularly those using compression garments to enhance performance and recovery, and in medical support appliances [20].

## 3. New Materials from Nature

### 3.1. Textiles from Food Waste

Many researchers have investigated the potential of using the waste from the food industry as raw material for textile production and the creation of more sustainable products. Using the unwanted skins and pulps of fruit from juice manufacturing has revealed a rich supply of raw material.

Companies such as Orange Fiber S.R.L. (Catania, Italy), who are utilizing some of the 700 million tonnes of waste from the Sicilian orange juice industry, are creating what is claimed to be a sustainable fibre. Cellulose is extracted from the waste, and is then processed and spun into useable yarn for both knitted and woven fabric production. The company also claims that additional benefits can be gained from the product. It has been well documented that extracts of citrus fruit peel can contain compounds such as essential oils, natural colors, and phenols, which all have associated biological activities such as antioxidant, antimicrobial and anti-inflammatory effects [21]. Orange Fiber is particularly marketing the additional skin moisturizing benefits of their textile, claiming this is due to embedded essential oils [22]. The process is currently filed for patent [23]. 

However, the true sustainable nature of the fiber is to be questioned. While the raw material would otherwise be waste, the extraction of the cellulose and its manufacture into useable textiles requires several stages of processing, some of which involve chemicals such as hydrogen peroxide, that could be deemed harmful to the environment [24]. Additionally, the performance characteristics of the end textile are not clearly documented and may need some development. The process, detailed in the patent application, suggests that the orange extract is blended with cellulose fibers extracted from wood pulp [23]. The extraction of cellulose from wood pulp is an established route for textile production and is available in the mass market under trade names such as Tencel™ [25].

A different approach to waste utilization is to take the leaves of the plant and use this as fiber to create textiles. Pinatex^®^ uses this principle, taking waste pineapple leaves from food production in the Philippines and processing these into fiber via a manufacturing unit in Spain. The leaves are stripped of excess surface biomatter (decortified) leaving the fibrous inner [26], with any waste from this process being used as biofuel. The fibers then undergo standard textile bulk manufacturing techniques to create rolls of fabric in a nonwoven textile structure, similar to felt. The nonwoven fabric is then chemically treated using established textile finishing processes to give the resulting material a leather-/canvas-like appearance and feel. The resultant textile can be made thicker or thinner to suit the end requirement. It can be coated to enhance durability and performance characteristics such as waterproofness or strength. However, while the fiber itself is 100% biodegradable, the finishing and coating chemicals that are used to make the resultant textile useable are not biodegradable, they are petroleum-based [27]. This is an area that needs to be resolved to meet the demands of a completely sustainable product. Nevertheless, the product is extremely versatile and is being used for a range of commercially available high-end fashion applications where sustainability is used as a marketing focus, such as shoes (Hugo Boss, Metzingen, Germany) and handmade bags (Artesano, Miami, FL, USA) [28]. The product has also been used for clothing, however, these are predominantly signature pieces from independent designers [29,30]; therefore, the ability of the garments to be subjected to standard consumer wash and wear demands should be questioned. Additionally, Pinatex^®^ fabrics have been used in automotive upholstery [27], albeit a concept product only.

One of the largest sources of waste in the food industry is that of seafood shells. Once the edible part of the animal has been prepared and eaten, the shell is discarded. While the shells do biodegrade, and are not contributing to ever increasing landfill, the exoskeleton structure is now seen as an untapped resource. Chitin is the long-chain polysaccharide that is found in the shells; chitosan is created by the deacetylation of chitin. Chitosan has previously found applications in the biomedical and agricultural industries. It can also be spun into fiber using solution spinning techniques, and has been explored as a textile for garment use. While not suitable in its pure form, it has been blended with viscose by Swicofil AG (Emmen, Switzerland), a commercial specialty fiber and yarn manufacturer, to create Crabyon© [31]. The resultant yarn can be processed into knitted or woven fabric structures, is extremely soft, and absorbs dyes readily. Additionally, the manufacturing process has been developed such that harmful organic solvents are not required, thus improving its eco-friendly status and preserving the natural biodegradability of the chitosan. The performance of the fabric allows it to be used in a range of commercially available garment applications from underwear and lingerie, to sportswear and school uniforms. The range of garment end-uses highlights the durability of the resultant product to domestic wash and wear processes.

### 3.2. Mushrooms

Mycelium has been explored for some time in the packaging industry, with companies such as Ecovative Design LLC (Green Island, NY, USA) leading the market in this innovation [32]. The technology uses the mycelium of the mushroom and grows these under controlled conditions to produce three-dimensional (3D) packaging structures. This principle has been applied to the more traditional textile fabric structure, with companies such as MycoWorks (San Francisco, CA, USA) leading in the field [33]. The company uses the growing process of mycelium to bind with organic matter, thus creating a “solid” textile, which is more akin to leather in appearance, rather than a traditional knit or woven fabric. The resulting material is flexible, durable, can be dyed easily and with natural dyestuffs, and has a degree of water repellency. It can be grown to shape, thus minimizing any waste. The mycelium structure is biodegradable, but the process is only initiated when the material comes into contact with soil-based bacteria. Therefore, it is suitable for various apparel and wider indoor textile applications [33]. Due to the ability to grow the mycelium-based material into 3D forms, this material has also found applications in furniture and architectural structures [34]. 

### 3.3. Microbes

Bacterial cellulose, a relation to plant cellulose or cotton, is one of the most abundant, naturally occurring polymers on the planet. It is most prolifically produced by the *Gluconacetobacter* bacteria species, which can be most commonly found in rotting fruit [35,36,37]. In contrast to plant cellulose, bacterial production yields cellulose in a highly crystalline state, imparting properties such as improved tensile strength and enhanced water absorbency [35,38]. Additionally, bacterial cellulose occurs in a highly pure form of the polymer, eliminating the need to heavily process the material before it can be subjected to further treatments, such as dyeing, which is the case with plant cellulose. Bacterial cellulose has found applications in the food, paper, and medical industries [39]. However, it has yet to break through into the mainstream fashion arena, despite applications being explored in synthetic fiber coatings and nonwoven cloths [40,41].

Fashion design researchers, such as Suzanne Lee, have experimented with kombucha [42], a symbiotic culture of bacteria and yeast (SCOBY), which is used to ferment tea and is purported to have health-giving properties. When brewed for extended periods, a biofilm forms on the surface of the tea liquor (Figure 2) made up of bacterial cellulose nanofibrils. This biofilm (or mat), when removed from the liquor, and then rinsed and dried, has aesthetic and physical properties like those of fine animal leather. It is commonly referred to as “vegetable leather”, however, under high magnification, the structure resembles that of a nonwoven fabric (Figure 3). Furthermore, the biofilm takes the shape of the container in which it is grown, so while whole mats of textile are produced, there is the potential to grow the mats to the shape required for the end product, thus addressing the issue of excess waste fabric in traditional textile manufacturing processes [3]. Conceptual garments have been produced to illustrate the potential of the vegetable leather to be subjected to garment construction techniques such as stitching, bonding and forming into 3D shapes. However, the extremely hydrophilic nature of the bacterial cellulose means it is not suitable for wear in conditions where there may be increased humidity, such as next to the human skin, and therefore, cannot be subjected to domestic washing. Despite this, research continues in this field as the “vegetable leather” can be composted at the end of its useful life and is seen as a viable proposition to alleviate the issue of fabric going to landfill sites.

## 4. Conclusions

There is no doubt that the textile industry is in crisis and the need for radical change is required. Textiles from natural sources are often assumed as eco-friendly or sustainable, but the evolution of both crop and animal farming has led to extreme demands on land, often with devastating effects. The advancement of synthetic textiles has also impacted the environment, with many of the textiles being nonbiodegradable and adding to landfill growth. 

While the textile industry has historically been influenced by nature (with developments such as Velcro^®^), these have generally been the adaptation of structural principles rather than using alternative natural resources. With consumer focus on minimizing waste, the use of food industry byproducts to create textiles is pertinent and shows promise. True alternative sources, such as mycelium and bacterial cellulose, are still in the early stages of development, but show potential in terms of textile performance properties.

However, textile manufacturers must create profit in order to be viable businesses. There is no doubt that product development costs money, and large investments are required for the investigation of new materials, particularly in such an innovative field as biomimetics. As is the case with Velcro ^®^, first introduced to the market over seventy years ago, these developments can be highly successful and profitable. Large, well-established, global businesses such as Swicofil AG and Schoeller Textil AG generally already have a steady product offering that generates income to support such developments and can explore relatively unknown fields. In the case of Schoeller Textil AG specifically, this has proved to be extremely successful as they are constantly broadening their offer of bulk-produced, bioinspired textiles. However, for smaller business where the innovative material is their only product, the future is more uncertain. These companies often rely on business and innovation awards to first become established, and are then often reliant on news or social media to promote their product and fuel further interest. They can also lack ease of access to established manufacturing equipment and expertise, which can prove prohibitive to their growth. Often, the best route to larger scale production for these innovations (such as MMT Textiles Inotek^TM^) is to patent the idea, then look for a more established manufacturer to develop a market and establish a viable bulk manufacturing process.

One must be also mindful that if any of these new sources are to become commercially viable, they must be able to perform to the standard required by the consumer. In apparel and fashion, comfort, flexibility, durability, and the potential to be washed are all key drivers. It is well established in the textile industry that many of these properties can be imparted and enhanced by synthetic chemical finishes. However, this impacts the environment and diminishes the validity of the novel material. True natural alternatives must be engineered to ensure they meet this criterion if they are to present themselves as viable substitutes.

## Figures and Tables

**Figure 1 biomimetics-04-00016-f001:**
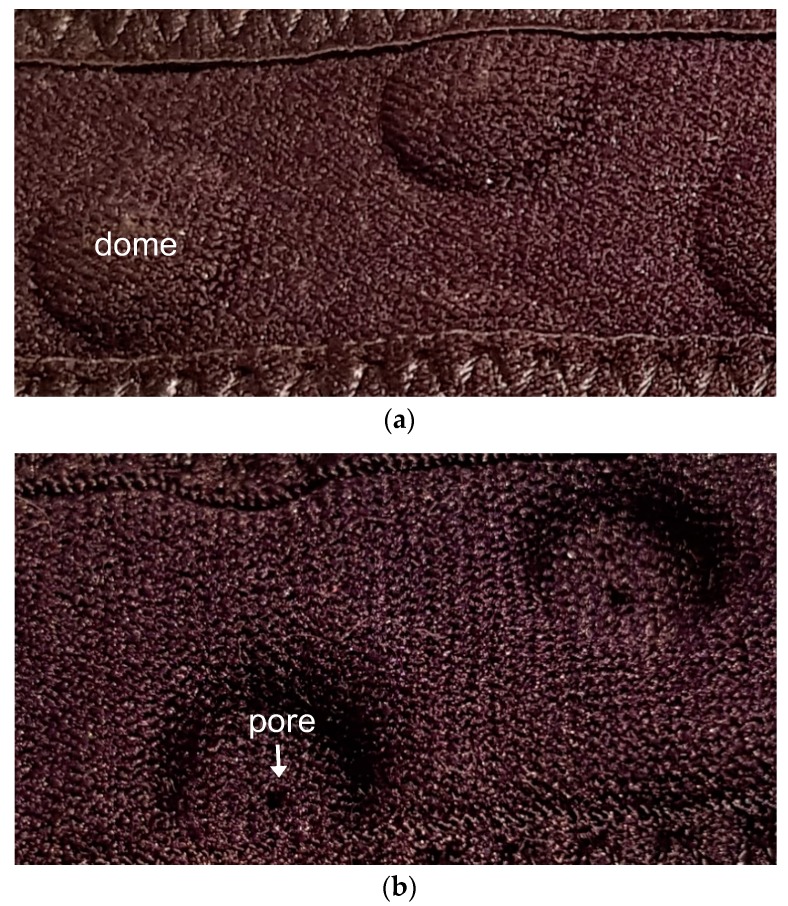
(**a**) Front view and (**b**) reverse view of Stomatex^®^ fabric showing the fabric domes and the pore at the dome apex.

**Figure 2 biomimetics-04-00016-f002:**
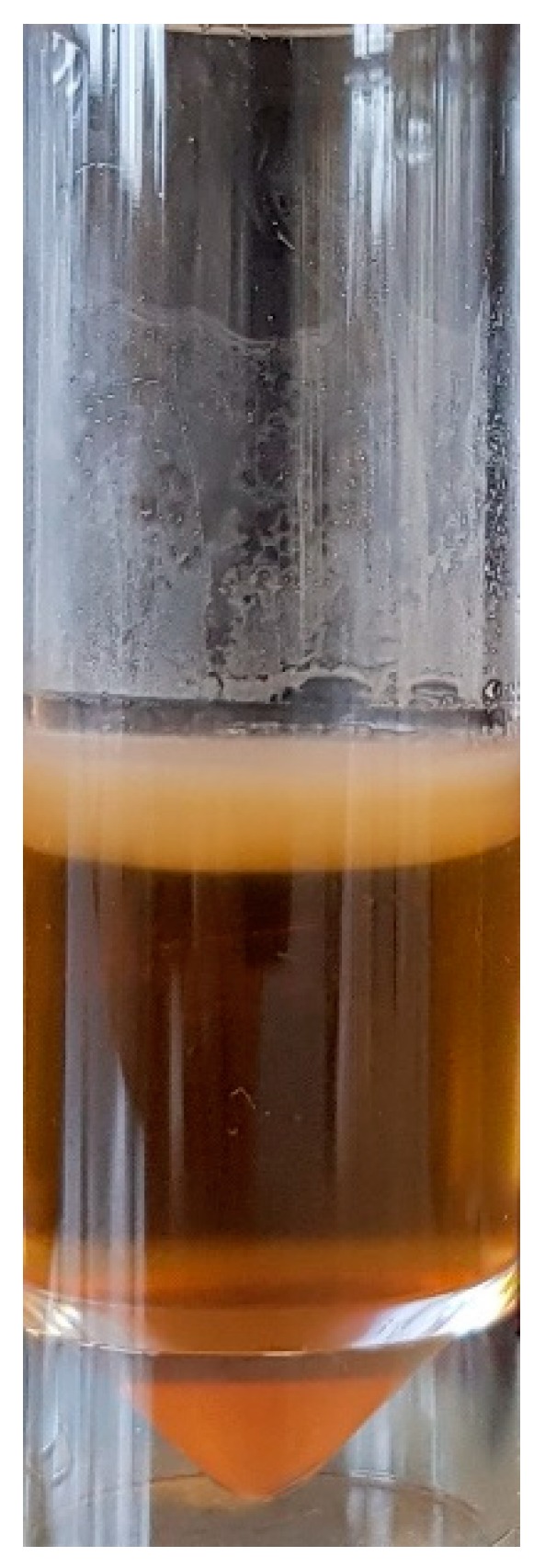
A bacterial cellulose “biofilm” being formed on the surface of a liquid medium.

**Figure 3 biomimetics-04-00016-f003:**
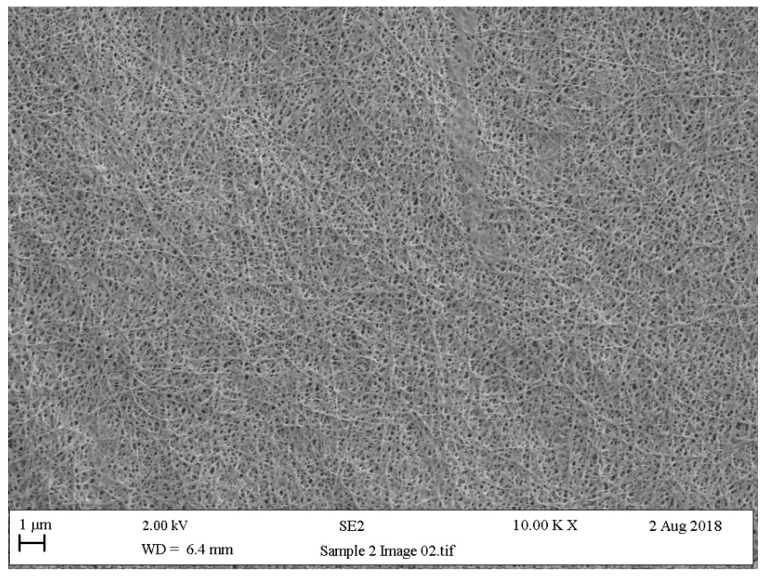
Scanning electron microscopy image of a dried bacterial cellulose biofilm (10,000× magnification).

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
