# Peer review of "Bioinspiration in Fashion—A Review"

_biomimetics, 2019, doi:10.3390/biomimetics4010016_

Reviewer 1 Report

I appreciate the time and effort that goes into preparing a review, especially with the number of references you cite. I enjoyed the paper. One sentence did cause me pause however: 

In contrast to plant cellulose, 158 bacterial production yields cellulose in a highly crystalline state imparting enhance properties such 159 as improved tensile strength and enhanced water absorbency [25]. 

In general, enhanced crystallinity in polymers results in lower accessibility, so enhanced water absorbency is a surprise. Are you sure the reference you cite got it right?

Author Response

Many thanks for taking the time to review my paper and thankyou for your kind and supportive comments. I have addressed these in the script and have provided additional details below:

I appreciate the time and effort that goes into preparing a review, especially with the number of references you cite. I enjoyed the paper. One sentence did cause me pause however: 

In contrast to plant cellulose, 158 bacterial production yields cellulose in a highly crystalline state imparting enhance properties such 159 as improved tensile strength and enhanced water absorbency [25]. 

In general, enhanced crystallinity in polymers results in lower accessibility, so enhanced water absorbency is a surprise. Are you sure the reference you cite got it right?

This is an area in which I have read extensively - and your comment did cause me to double check my findings. There are several papers that have undertaken work to explore the feature of enhanced absorbency and I have have added an additional reference to my script to further clarify this.

Reviewer 2 Report

Wood presents a brief review on (both traditional and alternative) technologies being used in the textile industry. The review begins with discussing bio-inspired design solutions (Velcro, pinecones, sharkskin) then discusses new materials from nature that can be employed in textiles. While the concept in general is of interest and the topic is certainly of global relevance, the review is missing a significant level of detail and depth. The following points should be addressed/added prior to publication:

-          There are no figures in this review! The article would benefit significantly from figures. The author should consider incorporating at least 2-3 figures that would help guide the reader through their manuscript.

-          Each subsection would benefit from more description and depth.  Within any single subsection, the texts seem quite disparate with minimal connectivity (ex, Pine cone section from Schoeller Textil to Inotek.) Furthermore, it is not clear how the cellulose fibers from Inotek relate to the Pinecone inspiration. The author should be more explicit in their writing especially in making and drawing conclusions based on weakly cited sections.

-          There is no discussion on the materials used (polymers, composites, structures, patterns, weaves, threads) in the textiles that are designed to mimic natural systems. Consider adding more technical detail in the writing of section #2. Consider adding these details.

-          The author makes no reference to how natural materials are processed into generating new textiles. For instance, how does this work? Is it feasible for bulk manufacturing (similar to production levels of cotton or silk – another natural material which never gets discussed)? How durable are these materials? Can they withstand abrasion/wash tests? What about multiple wear cycles? Or are these just one-off pieces?

-          How successful/sustainable are the companies that are referenced throughout the text? It would be helpful to see a table with the companies’ names, the year they were founded, the source of their material; their annual profit. This would certainly raise the impact and quality of the review

Author Response

Many thanks for taking the time to review my manuscript and for your valuable and helpful comments. I have amended the paper accordingly and have addressed each of your points as detailed below. I hope you find all these to your satisfaction:

There are no figures in this review! The article would benefit significantly from figures. The author should consider incorporating at least 2-3 figures that would help guide the reader through their manuscript.

Figures have now been added to the manuscript to illustrate some of the points.

Each subsection would benefit from more description and depth.  Within any single subsection, the texts seem quite disparate with minimal connectivity (ex, Pine cone section from Schoeller Textil to Inotek.) Furthermore, it is not clear how the cellulose fibers from Inotek relate to the Pinecone inspiration. The author should be more explicit in their writing especially in making and drawing conclusions based on weakly cited sections.

I have revised each section in line with the above with the aim of being more explicit in the description, particularly within the pine cone section. I agree with the comment that this was not clearly addressed in the original document and I have explained the links more clearly in the revised script. I have also cited more widely throughout.

There is no discussion on the materials used (polymers, composites, structures, patterns, weaves, threads) in the textiles that are designed to mimic natural systems. Consider adding more technical detail in the writing of section #2. Consider adding these details.

Thankyou for highlighting this point. I am in agreement that in terms of technical textile processing and manufacturing terminology, the original script was lacking. However, I was mindful that I was not writing for a textile specific journal and therefore tried to avoid too much textile specific detail. However, I do agree that in order to explore the relevance of the subject matter for the textile industry there is benefit to adding more textile processing information. I have done this and tried to explore the points to gain relevance, whilst being mindful of the audience. I have also referenced textile texts more widely to support the writing.

The author makes no reference to how natural materials are processed into generating new textiles. For instance, how does this work? Is it feasible for bulk manufacturing (similar to production levels of cotton or silk – another natural material which never gets discussed)? How durable are these materials? Can they withstand abrasion/wash tests? What about multiple wear cycles? Or are these just one-off pieces?

Again, thankyou for this and again, I agree, this needed to be explored more deeply. For each textile area I have added detail around manufacturing processes and how this links with the wash and wear performance expected by the consumer for everyday wear. For example, I have highlighted clearly in the Pinatex section that some of the products are concept pieces, whilst in the Swicofil Crabyon section I have commented the versatility and durability of the product illustrated by its range of end uses.

Whilst I have made reference to cotton in the introduction, it was a conscious decision not to include current natural products such as cotton and silk as I wanted the review to focus on innovations around biomimicry and new fibre sources, rather than currently established natural fibres.

How successful/sustainable are the companies that are referenced throughout the text? It would be helpful to see a table with the companies’ names, the year they were founded, the source of their material; their annual profit. This would certainly raise the impact and quality of the review

I completely agree, this would add some real depth and I thank you for this useful observation. However, to address this is more difficult than it may seem. Many of the larger companies mentioned in the script (e.g. Schoeller and Swicofil) are subsidiaries of larger organisations and as such, it is difficult to find the figures relating the the specific business. Moreover, due to the diversity of the product range of these larger organisations, I think it may be tricky to attribute any changes in fortune to specific products. 

Conversely, at the other end of the spectrum, some of the smaller establishments, such as MMT or Orange Fiber have only one product which is in its infancy and have therefore little to report in terms of company financial statistics.

All that said, it is a valuable point that needs to be mentioned and I have added a section in the conclusion to address this which I hope you will find satisfactory.

Round  2

Reviewer 2 Report

The author has addressed many of the original concerns. The paper is now suitable for publication.